nanotechnology/computational biology

supervised learning algorithms, skin tissue engineering, polymeric scaffold performance, cell–material interaction, predictive model

**Author for correspondence:**
Archana Bhaw-Luximon
e-mail: abluximon@gmail.com,
a.luximon@uom.ac.mu

This article has been edited by the Royal Society of Chemistry, including the commissioning, peer review process and editorial aspects up to the point of acceptance.

# Correlating *in vitro* performance with physico-chemical characteristics of nanofibrous scaffolds for skin tissue engineering using supervised machine learning algorithms

Lakshmi Y. Sujeeun[1,2], Nowsheen Goonoo[1], Honita Ramphul[1], Itisha Chummun[1], Fanny Gimié[3], Shakuntala Baichoo[2] and Archana Bhaw-Luximon[1]

[1]Biomaterials, Drug Delivery and Nanotechnology Unit, Centre for Biomedical and Biomaterials Research (CBBR), and [2]Department of Digital Technologies, Faculty of Information, Communication and Digital Technologies, University of Mauritius, 80837 Réduit, Mauritius
[3]Animalerie, Plateforme de recherche CYROI, 2 rue Maxime Rivière, 97490 Sainte Clotilde, Ile de La Réunion, France

LYS, 0000-0002-0030-2135; NG, 0000-0002-1626-7282;
IC, 0000-0001-5511-7140; SB, 0000-0002-9335-1939;
AB-L, 0000-0001-6215-6420

The engineering of polymeric scaffolds for tissue regeneration has known a phenomenal growth during the past decades as materials scientists seek to understand cell biology and cell–material behaviour. Statistical methods are being applied to physico-chemical properties of polymeric scaffolds for tissue engineering (TE) to guide through the complexity of experimental conditions. We have attempted using experimental *in vitro* data and physico-chemical data of electrospun polymeric scaffolds, tested for skin TE, to model scaffold performance using machine learning (ML) approach. Fibre diameter, pore diameter, water contact angle and Young's modulus were used to find a correlation with 3-(4,5-dimethylthiazol-2-yl)-2,5-diphenyltetrazolium bromide (MTT) assay of L929 fibroblasts cells on the scaffolds after 7 days. Six supervised learning algorithms were trained on the data using Seaborn/Scikit-learn Python libraries. After hyperparameter tuning, random forest regression yielded the highest accuracy of 62.74%. The predictive model was also correlated with *in vivo* data. This is a first preliminary study on ML methods for the prediction of cell–material interactions on nanofibrous scaffolds.

# 1. Introduction

Tissue engineering (TE) scaffolds are primarily engineered to: (i) stimulate cell–material interactions and adhesion, and extracellular matrix (ECM) deposition, (ii) allow adequate transport of biological factors to enable cell survival, proliferation and differentiation, and (iii) promote processes such as angiogenesis and reduce inflammation [1]. Scaffolds have been developed for the engineering of a range of tissues such as bone, skin, muscle, vascular, neural, cartilage and ligament [2]. Regardless of the tissue type, a number of key considerations are important when designing or determining the suitability of a scaffold for tissue regeneration, namely biocompatibility, biodegradability, sufficient mechanical strength to ensure integrity and interconnected porous structures and high porosity to allow cell penetration and tissue integration, vascularization for the transport of gases, diffusion of nutrient and growth factors [3–5].

Polymeric TE nanoscaffolds have experienced tremendous progress with a number of scaffolds being used in clinical setting. Materials scientists are now faced with more sophisticated challenges such as being able to match materials and scaffold design with wound healing requirements. For instance, a preliminary matching of *in vitro* data with scaffold's physico-chemical characteristics may offer a better comprehension of the cell behaviour and bring evidence-based data to scaffold design. The use of computational methods in three-dimensional printing techniques for scaffold design, fabrication and simulation has been studied [6]. To the best of our knowledge, no study in the literature, up to now, has specifically looked at modelling of cell–material behaviour on electrospun scaffolds. Scaffold material, structure and fabrication technique are the three main parameters which determine scaffold properties. Thus, designing a computational model which can integrate both material and structure data is a challenging task. It has been argued that machine learning (ML) can offer an indispensable tool and overcome challenges in the biomedical field involving complex heterogeneous data when conventional statistical tools have failed [7–10]. Predictive modelling is a probabilistic process that allows us to forecast outcomes, on the basis of predictors. The latter are features that come into play in the determination of the required result, i.e. the outcome of the model [11]. However, the huge potential of ML for predictive modelling in TE still remains mostly unexplored.

In this paper, we hypothesize that using *in vitro* cell culture data and ML approaches, we can reverse engineer scaffold performance and predict *in vivo* outcomes (scheme 1). We have used *in vitro* and physico-chemical characterization data, namely number of cells (quantified during 3-(4,5-dimethylthiazol-2-yl)-2,5-diphenyltetrazolium bromide (MTT) assay), fibre diameter, pore diameter, water contact angle and Young's modulus, generated from a series of experiments using biopolymer-based scaffolds, to build a predictive model for scaffold performance. Our goal is to predict the *in vitro* outcome, given the physico-chemical features. Six simplified regression models were used and their metrics were compared to determine the most accurate algorithm. The accuracy of the ML model was then assessed against *in vivo* experimental results.

# 2. Material and methods

## 2.1. Polymer blends

Polyhydroxybutyrate (PHB) (Sigma-Aldrich), poly(hydroxybutyrate-*co*-valerate) (PHBV) (HV content 12 mol%, Sigma-Aldrich), kappa-carrageenan (KCG) (Sigma-Aldrich), polydioxanone (PDX) (Resomer X 206S, inherent viscosity 2.0, Evonik), fucoidan (FUC) (Fucoidan from *Fucus vesiculosus* greater than or equal to 95%, Sigma-Aldrich), polysucrose (PSuc) and poly-L-lactic acid (PLLA) (PURASORB PL18, IV 1.8 g dl$^{-1}$, Purac) were used as purchased. Cellulose was extracted from locally available sugarcane bagasse using a combination of mercerization and bleaching treatments to generate an average % yield of 40 (±2). Cellulose acetate (CA) was synthesized from sugarcane bagasse-derived cellulose using an optimized acetylation method with an average % yield of 62 (±2). Nanosilica was generated through the acid hydrolysis of extracted silica (% yield 30 (±2)). The silica used for the preparation of nanosilica was extracted from sugarcane bagasse ash using the sol-gel process.

Thirteen scaffold families were included in this study: PHB/KCG, PHBV/KCG, PDX/FUC, PDX/KCG, PDX/PHBV, PDX/PSuc, PLLA/PSuc, PLLA/CA, PLLA/cellulose, PDX/CA, PLLA/CA 1% nanosilica, PLLA/cellulose 1% nanosilica and PDX/CA 1% nanosilica. Each family consisted of at least four blend compositions in triplicate results (of different polymer contents—100% of polymer A and 0% of polymer B; 90% of polymer A and 10% of polymer B; 80% of polymer A and 20% of polymer B; 70% of polymer A and 30% of polymer B; 60% of polymer A and 40% of polymer B; and 50% of polymer A and 50% of polymer B).

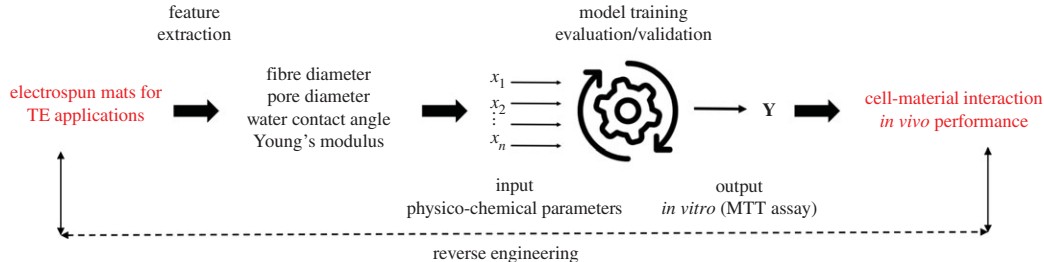

**Scheme 1.** Hypothesis.

**Table 1.** Summary of the mass of polymers and volume of solvents used for 80/20 blend ratio for various blend families.

| blend family | solution A | solution B |
|---|---|---|
| polyhydroxybutyrate (PHB)/kappa-carrageenan (KCG) | 400 mg of PHB in 8 ml of HFIP | 100 mg of KCG in 2 ml of CHCl$_3$ |
| poly(hydroxybutyrate-*co*-valerate) (PHBV)/KCG | 1200 mg of PHBV in 8 ml of HFIP | 300 mg of KCG in 2 ml of CHCl$_3$ |
| polydioxanone (PDX)/fucoidan (FUC) | 1400 mg of PDX in 9 ml of HFIP | 350 mg of FUC in 1 ml of DMF |
| PDX/KCG | 1200 mg of PDX in 7 ml of HFIP | 300 mg of KCG in 3 ml of CHCl$_3$ |
| PDX/PHBV | 800 mg of PDX and 200 mg of PHBV were dissolved in 10 ml of HFIP | |
| PDX/polysucrose (PSuc) | 960 mg of PDX and 240 mg of PSuc were dissolved in 10 ml of HFIP | |
| poly-L-lactic acid (PLLA)/PSuc | 800 mg of PLLA and 200 mg of PSuc were dissolved in 10 ml of HFIP | |
| PLLA/cellulose acetate (CA) | 800 mg of PLLA in 8 ml of HFIP | 200 mg of CA in 2 ml of HFIP |
| PDX/CA | 800 mg of PDX in 6.7 ml of HFIP | 200 mg of CA in 1.7 ml of HFIP |
| PLLA/CA 1% nanosilica | to a 80/20 solution of PLLA/cellulose acetate, 1% nanosilica (10 mg) was added and allowed to stir overnight | |
| PDX/CA 1% nanosilica | to a 80/20 solution of PDX/cellulose acetate, 1% nanosilica (10 mg) was added and allowed to stir overnight | |

## 2.2. Scaffold fabrication

Scaffolds were fabricated using the electrospinning method (bottom-up NE300 Laboratory scale electrospinner, Inovenso Company, Turkey). Parameters for electrospinning were varied depending on the polymers within the blend and on the blend composition so as to produce random bead-free fibres. PHB/KCG and PHBV/KCG fibres were produced as reported [12]. Electrospinning parameters for PSuc-based and cellulose-based fibres have been reported, respectively, by Chummun *et al.* [13] and Ramphul *et al.* [14]. The fabrication of PDX/KCG and PDX/FUC has been detailed in Goonoo *et al.* [15].

Most blend solutions were prepared by mixing two solutions (solution A and solution B) with the exception of PDX/PHBV, PDX/PSuc and PLLA/PSuc. Table 1 summarizes the mass of polymers as well as the volume of solvents used for the preparation of different blend solutions. PLLA/cellulose and PLLA/cellulose 1% nanosilica mats were obtained following the deacetylation of PLLA/CA and PLLA/CA 1% nanosilica mats. Briefly, the acetylated mats were immersed in 0.05 M NaOH solution for 48 h at room temperature followed by washing with distilled water.

## 2.3. Physico-chemical characterization

### 2.3.1. Fibre and pore diameters

The average fibre diameters of the electrospun mats were determined by scanning electron microscope (SEM) as reported previously [13,14,16]. Average pore diameters were determined by measuring the

diameter of at least 50 different pores using ImageJ software. Pores were identified as areas of void space bounded by fibres on all sides at or near the same depth of field. The shortest diagonal axes were measured and averaged together to obtain pore diameter. SEM images were taken using a Tescan Vega 3 LMU microscope (CBBR, Mauritius) with accelerating voltage of 30 kV.

### 2.3.2. Mechanical properties

A Universal Instron Tester 3344 (Instron, USA) was used to measure the tensile properties of the electrospun mats at 25°C. The electrospun mats cut in rectangular shapes ($4 \times 1$ cm) were clamped with gauge length set at 1 cm and strained at a rate of 10 mm min$^{-1}$ using a load cell of 2 kN.

### 2.3.3. Water contact angle

The contact angles of the electrospun mats were measured using Milli-Q water as a probe liquid with a Krüss Drop Shape Analyzer DSA 25 (Advanced Lab GmbH, Germany). Static contact angle data based on the sessile drop method were acquired immediately after deposition of a 2 µl drop on at least three positions for each sample and are stated as the arithmetic mean.

## 2.4. *In vitro* evaluation by MTT assay

The MTT assay was conducted using L929 mouse fibroblasts (Sigma-Aldrich, ECACC certified) after 7 days as reported earlier [12] and absorbance values were measured at a wavelength of 540 nm using a Thermo Scientific Varioskan LUX Multimode Microplate Reader.

## 2.5. *In vivo* biocompatibility tests

Surgical procedures were approved by the Animal Ethics Committee of CYROI, La Réunion, France (APAFIS Number: 2018052311219125V3). Wistar albino rats (8–10 weeks old), both males (308–407 g) and females (189–252 g), were used for *in vivo* studies ($n = 5$). Prior to implantation, the scaffolds were disinfected in 70% ethanol solution overnight. The rats were anaesthetized by isoflurane inhalation and the dorsal region was shaved followed by 1–2 cm dorsal incisions (two on each side of the spine). All the scaffolds were implanted as stacks of three each with dimensions $0.5 \times 1$ cm. The rats were provided with regular supply of food and water throughout the study period. They were also monitored daily for any signs of inflammation. The scaffolds were removed and the neighbouring skin tissues were harvested for histology analysis after two and four weeks, respectively. Histological analysis of tissues was performed using Masson's Trichrome staining. The stained slides were examined using light microscopy (Nanozoomer 560 digital slide scanner, Hamamatsu). For quantitative analysis of angiogenesis, the number of blood vessels were counted within a fixed distance of 100 µm from the explanted scaffold and the results were expressed as the number of blood vessels per mm$^2$ for each scaffold [17].

## 2.6. Data preparation and supervised machine learning procedure

Before applying statistical learning to scaffold data, standard pre-processing was performed. Data were collected, cleaned and rearranged in a format appropriate for meaningful statistical analyses. The dataset used for this study contained 182 observations, four numeric features characterizing the scaffolds—fibre diameter, pore diameter, water contact angle and Young's modulus—and the target variable: number of cells. The variables were scaled by the MinMaxScaler method in order to bring the data on a relatively similar scale and close to the normal distribution. Low variance filter and high correlation filter using correlation matrix with Pearson correlation were applied on the features. Feature selection was performed with random forest regressor, recursive feature elimination (RFE) and forward feature selection (FFS) [18].

For predictive modelling settings, a data matrix denoted $X$ is needed, and $y$ as the target variable (to predict). We used 80% of the data for training, and the remaining 20% was used for testing. Six regression algorithms, namely linear regression, support vector regression (SVR), random forest regression, lasso regression, decision tree regression and k-nearest neighbour (k-NN) regression were applied to the data and compared to obtain the best performance. The six ML algorithms have undergone a training process to find patterns in the training data that map the input parameters to the target. Each algorithm learned from the training dataset containing both inputs and outputs to generate a ML

model. Each ML model was then tested using the test dataset. This metric has allowed us to evaluate how the model might perform against unseen data. To optimize the training phase, hyperparameter tuning was performed for each model in order to improve its accuracy. All codes were implemented in Python 3 using Seaborn/Scikit-learn libraries [19].

### 2.6.1. Importing Python libraries and loading the dataset into a data frame

Required libraries numpy and pandas were imported, and the read_csv method of pandas was used to load the dataset 'scaffold_data.csv' into a pandas data frame *df*, as shown below.

```
import pandas as pd
import numpy as np
df = pd.read_csv('scaffold_data.csv')
```

### 2.6.2. Splitting the dataset into training and test sets

Before feeding data to a regression model, some pre-processing is needed. A variable $X$ was created from the initial dataset *df* to contain the four scaffold parameters (i.e. features): fibre diameter, pore diameter, water contact angle and Young's modulus. A variable $y$ was created to represent the target variable, number of cells. The train_test_split function of Scikit-learn was used to split the data into training and test sets. test_size = 0.2 indicates that 20% of the initial dataset was used for testing, and 80% for training. random_state ensures reproducibility and X_train, X_test, y_train and y_test were defined for the ouput of train_test_split.

```
from sklearn.model_selection import train_test_split
X = df['fibre_diameter','pore_diameter','contact_angle','youngs_modulus']
Y = df['nb_cells']
X_train, X_test, y_train, y_test = train_test_split(X, y, test_size = 0.2,
random_state = 0)
```

As variance is range dependent, feature scaling is required on the variable $X$ before training the data. The MinMaxScaler method was used to scale and translate each feature individually such that it is in the given range on the training set, i.e. between zero and one.

```
from sklearn.preprocessing import MinMaxScaler
scaler = MinMaxScaler()
scaler.fit(X)
X_train = scaler.transform(X_train)
X_test = scaler.transform(X_test)
```

### 2.6.3. Creating a random forest regression model and fitting it to the training data

X_train and y_train obtained above were used to train a random forest model from RandomForestRegressor. The fit method was used to pass the parameters defined below. The output of this step describes a large number of parameters for the random forest model. These parameters are tuneable, and can be adjusted to optimize the accuracy of the model.

```
from sklearn.ensemble import RandomForestRegressor
model = RandomForestRegressor(n_estimators = 100, random_state = 0,
max_depth =4)
model.fit(X_train, y_train)
```

### 2.6.4. Prediction

Once the model is trained, it is ready to make predictions. The predict method was used on the model and X_test was passed as a parameter to get the output as y_pred.

```
y_pred=model.predict(X_test)
y_pred
```

## 2.7. Statistical analysis

All group data were reported as the arithmetic mean ± standard error of mean. Statistical analysis was performed with GraphPad Prism software. Statistical differences were analysed using unpaired Student $t$-test. Differences were considered statistically significant for $p < 0.05$ (*significant at $p < 0.05$, **significant at $p < 0.01$, ***significant at $p < 0.001$, ****significant at $p < 0.0001$). Regression metrics such as accuracy measure, Spearman correlation and mean absolute percentage error (MAPE) were computed to evaluate the predictive accuracy of each ML model. Correlation between the actual and predicted output data was performed to assess the adequacy of each ML model.

# 3. Results and discussion

## 3.1. Physico-chemical characterization of nanoscaffolds: fibre diameter, pore diameter, water contact angle and Young's modulus

In this study, we have used nanofibrous scaffolds generated through the electrospinning process. We have previously reported on the synthesis of these scaffolds which have been assessed for skin TE both *in vitro* and *in vivo* [12–15,17]. These scaffolds are engineered using blend solutions of various biopolymer combinations (table 2) and we have shown that the cell-scaffold response depended on the combination of materials as well as the physico-chemical properties of the scaffolds namely pore diameter, fibre diameter, water contact angle and Young's modulus (table 2). Fibre diameter and pore diameter were determined using ImageJ software. The data are presented as arithmetic mean ± standard error of mean. The different combinations of biomaterials used in this study will allow the testing of the robustness of the ML models to relate biological data with the physico-chemical characteristics of the scaffolds.

L929 fibroblasts cells were grown on the scaffolds and cell proliferation was assessed using MTT after 7 days. Fibroblasts were chosen due to their crucial role in tissue regeneration in general across all tissue types. They are responsible for the proliferative phase in skin TE and are the major producer of ECM for the growth of keratinocytes and endothelial cells. Fibroblasts and macrophages are the first cells which interact with the surface of TE scaffolds in wounds and thus, determine the fate of the scaffolds as an efficient medical device.

Our experimental work indicated that there is no linear relationship between physico-chemical parameters and *in vitro* results. For instance, an increase in pore diameter did not translate directly to an increase in cell number but would influence contact angle and mechanical property, and together these will regulate cell–material interaction. The right balance between these properties needs to be determined to predict scaffold performance *in vitro*. Another observation that we made concerned the fibroblast morphology and behaviour (figure 1). For instance, changing the biopolymer from KCG to PSuc to cellulose caused a change in L929 morphology from elongated to dendritic to flat dense clusters, respectively. Each morphology change is an indication of a different type of cell–material interaction. Thus, feature selection was performed to select the predominant physico-chemical parameter which will impact not only cell behaviour but also the other physico-chemical parameters.

The main feature of ML is learning from experience. In the first step of supervised learning, a labelled training input dataset is fed to a ML algorithm. With the training dataset, the system adjusts itself by modifying parameters to create a logical model. The performance of the built model is then tested with a test dataset which the model has never seen before and the accuracy of the ML model is evaluated. Six supervised ML algorithms were used to predict scaffold performance with *in vitro* cell culture data.

**Table 2.** Physico-chemical characteristics of scaffolds and *in vitro* MTT assay of L929 fibroblasts after 7 days.

| nanoscaffolds | pore diameter (μm) | fibre diameter (μm) | water contact angle (°) | Young's modulus (MPa) | number of cells as determined from MTT assay |
|---|---|---|---|---|---|
| polyhydroxybutyrate (PHB)/kappa-carrageenan (KCG) | | | | | |
| 100/0 | 1.9 ± 0.7 | 1.3 ± 0.4 | 126 ± 1 | 518.2 ± 36.5 | 56 068 ± 3498 |
| 90/10 | 1.2 ± 0.6 | 0.9 ± 0.5 | 120 ± 1.9 | 271.7 ± 58.9 | 49 515 ± 10 347 |
| 80/20 | 0.9 ± 0.3 | 0.7 ± 0.5 | 107 ± 1.3 | 160.5 ± 32.8 | 35 661 ± 3079 |
| 70/30 | 1.1 ± 0.5 | 0.7 ± 0.5 | 104 ± 0.9 | 90.6 ± 10.9 | 38 083 ± 11 082 |
| poly(hydroxybutyrate-*co*-valerate) (PHBV)/KCG | | | | | |
| 100/0 | 0.8 ± 0.3 | 0.4 ± 0.1 | 112 ± 1 | 197.2 ± 44.6 | 73 421 ± 3834 |
| 90/10 | 1.0 ± 0.5 | 0.7 ± 0.2 | 73.6 ± 2.8 | 160 ± 14.8 | 59 800 ± 17 584 |
| 80/20 | 0.9 ± 0.3 | 0.6 ± 0.2 | 62.4 ± 0.5 | 111.7 ± 22.2 | 45 060 ± 12 110 |
| 70/30 | 1.0 ± 0.5 | 0.5 ± 0.2 | 57.8 ± 1.7 | 108.5 ± 6.8 | 46 109 ± 10 857 |
| polydioxanone (PDX)/fucoidan (FUC) | | | | | |
| 100/0 | 0.2 ± 0.06 | 0.3 ± 0.1 | 32.1 ± 0.0 | 73.8 ± 7.6 | 32 372 ± 6793 |
| 90/10 | 0.2 ± 0.08 | 0.2 ± 0.07 | 32.1 ± 0.0 | 69.6 ± 8.4 | 7510 ± 4328 |
| 80/20 | 0.2 ± 0.1 | 0.2 ± 0.05 | 32.1 ± 0.0 | 38.1 ± 3.9 | 13 691 ± 4154 |
| 70/30 | 0.2 ± 0.1 | 0.2 ± 0.05 | 32.1 ± 0.0 | 35 ± 11.3 | 21 154 ± 6873 |
| PDX/KCG | | | | | |
| 100/0 | 2.0 ± 0.7 | 1.1 ± 0.3 | 32.1 ± 0.0 | 73.8 ± 7.6 | 32 372 ± 6793 |
| 90/10 | 1.6 ± 0.5 | 1.0 ± 0.2 | 32.1 ± 0.0 | 72 ± 6.5 | 21 831 ± 6967 |
| 80/20 | 1.5 ± 0.6 | 0.9 ± 0.2 | 32.1 ± 0.0 | 42.6 ± 6.3 | 16 606 ± 3756 |
| 70/30 | 0.9 ± 0.3 | 0.5 ± 0.2 | 32.1 ± 0.0 | 38.2 ± 5.5 | 30 576 ± 3825 |
| PDX/PHBV | | | | | |
| 100/0 | 1.1 ± 0.4 | 0.4 ± 0.1 | 32.1 ± 0.0 | 73.8 ± 7.6 | 24 956 ± 1969 |
| 90/10 | 1.7 ± 0.6 | 1.0 ± 0.3 | 32.1 ± 0.0 | 95.6 ± 11.6 | 25 072 ± 4328 |
| 80/20 | 1.2 ± 0.4 | 0.7 ± 0.3 | 105.1 ± 2.2 | 72.9 ± 6.9 | 23 136 ± 6132 |
| 70/30 | 1.2 ± 0.5 | 0.6 ± 0.2 | 119.6 ± 2.5 | 100.4 ± 18.3 | 25 912 ± 9796 |
| PDX/PSuc | | | | | |
| 100/0 | 8.9 ± 5.4 | 1.0 ± 0.04 | 32.1 ± 0.0 | 51.8 ± 10.3 | 35 708 ± 6551 |
| 90/10 | 6.2 ± 4.2 | 0.8 ± 0.1 | 32.1 ± 0.0 | 43.0 ± 10.4 | 33 721 ± 17 362 |
| 80/20 | 5.1 ± 2.8 | 0.8 ± 0.03 | 32.1 ± 0.0 | 31.5 ± 5.2 | 57 370 ± 31 770 |
| 70/30 | 4.1 ± 3.2 | 0.7 ± 0.06 | 32.1 ± 0.0 | 83.2 ± 23.9 | 41 445 ± 14 515 |
| 60/40 | 3.6 ± 2.3 | 0.7 ± 0.03 | 32.1 ± 0.0 | 58.2 ± 27.5 | 10 1115 ± 73 374 |
| 50/50 | 3.4 ± 2.5 | 0.6 ± 0.04 | 32.1 ± 0.0 | 33.0 ± 2.0 | 18 131 ± 13 203 |
| PLLA/PSuc | | | | | |
| 100/0 | 5.7 ± 3.3 | 1.0 ± 0.05 | 141.3 ± 2.0 | 235 ± 15 | 51 408 ± 17 250 |
| 90/10 | 4.2 ± 2.5 | 0.9 ± 0.1 | 135.1 ± 1.6 | 109 ± 8.7 | 53 312 ± 13 995 |
| 80/20 | 3.9 ± 2.4 | 0.8 ± 0.03 | 134.0 ± 0.7 | 114.3 ± 4.0 | 56 111 ± 14 113 |
| 70/30 | 2.9 ± 2.0 | 0.7 ± 0.04 | 126.3 ± 4.7 | 85.7 ± 5.9 | 51 717 ± 31 051 |
| 60/40 | 2.01 ± 0.8 | 0.7 ± 0.01 | 132.9 ± 1.1 | 90.7 ± 5.5 | 49 338 ± 10 401 |
| 50/50 | 2.4 ± 1.5 | 0.6 ± 0.02 | 81.6 ± 9.5 | 99.3 ± 31.6 | 41 473 ± 12 501 |

*(Continued.)*

**Table 2.** (*Continued.*)

| nanoscaffolds | pore diameter (µm) | fibre diameter (µm) | water contact angle (°) | Young's modulus (MPa) | number of cells as determined from MTT assay |
|---|---|---|---|---|---|
| PLLA/CA | | | | | |
| 0/100 | 0.83 ± 0.3 | 0.7 ± 0.08 | 134.4 ± 1.8 | 90.4 ± 15.3 | 10 043 ± 6267 |
| 100/0 | 5.7 ± 3.3 | 1.0 ± 0.3 | 140.7 ± 1.8 | 235.4 ± 14.8 | 51 408 ± 17 450 |
| 90/10 | 2.3 ± 0.4 | 0.9 ± 0.3 | 138.4 ± 0.8 | 251.8 ± 23.4 | 53 228 ± 25 665 |
| 80/20 | 2.7 ± 0.5 | 0.8 ± 0.4 | 139.6 ± 0.7 | 173.3 ± 29.2 | 18 719 ± 4141 |
| 70/30 | 3.5 ± 1.1 | 0.6 ± 0.2 | 119.5 ± 0.9 | 58.9 ± 3.3 | 17 040 ± 7452 |
| 60/40 | 2.5 ± 0.7 | 0.5 ± 0.3 | 127.7 ± 0.7 | 41.2 ± 17.6 | 22 442 ± 5173 |
| 50/50 | 2.0 ± 0.6 | 0.5 ± 0.2 | 117.3 ± 1.3 | 104.4 ± 13.8 | 7552 ± 3618 |
| PLLA/cellulose | | | | | |
| 0/100 | 1.5 ± 0.6 | 0.3 ± 0.1 | 25.0 ± 0.0 | 162.3 ± 21.3 | 31 678 ± 14 856 |
| 100/0 | 5.7 ± 3.3 | 1.0 ± 0.3 | 140.7 ± 1.8 | 235.4 ± 14.8 | 51 408 ± 17 450 |
| 90/10 | 2.3 ± 0.4 | 0.8 ± 0.3 | 97.2 ± 0.4 | 275.0 ± 36.8 | 34 028 ± 17 333 |
| 80/20 | 2.7 ± 0.3 | 0.7 ± 0.2 | 25.0 ± 0.0 | 146.5 ± 10.6 | 34 420 ± 20 453 |
| 70/30 | 3.5 ± 1.0 | 0.6 ± 0.2 | 25.0 ± 0.0 | 95.7 ± 6.7 | 33 161 ± 23 589 |
| 60/40 | 3.0 ± 0.5 | 0.4 ± 0.2 | 25.0 ± 0.0 | 27.6 ± 12.4 | 30 444 ± 21 740 |
| 50/50 | 2.3 ± 0.5 | 0.3 ± 0.07 | 25.0 ± 0.0 | 102.6 ± 72.8 | 32 237 ± 11 258 |
| PDX/CA | | | | | |
| 100/0 | 8.9 ± 5.4 | 1.2 ± 0.5 | 32.1 ± 0.0 | 51.8 ± 10.3 | 35 708 ± 6551 |
| 90/10 | 3.2 ± 1.5 | 1.0 ± 0.3 | 32.1 ± 0.0 | 45.8 ± 4.6 | 45 280 ± 3130 |
| 80/20 | 2.6 ± 1.0 | 1.0 ± 0.5 | 32.1 ± 0.0 | 90.9 ± 16.0 | 35 904 ± 3914 |
| 70/30 | 4.8 ± 2.1 | 0.8 ± 0.3 | 32.1 ± 0.0 | 134.5 ± 15.5 | 33 665 ± 24 790 |
| 60/40 | 1.7 ± 0.6 | 0.6 ± 0.3 | 32.1 ± 0.0 | 191.8 ± 24.0 | 16 620 ± 8480 |
| 50/50 | 2.2 ± 0.5 | 0.6 ± 0.2 | 32.1 ± 0.0 | 65.7 ± 14.0 | 30 446 ± 10 504 |
| PLLA/CA 1% nanosilica | | | | | |
| 100/0 | 5.7 ± 3.3 | 1.0 ± 0.1 | 128.6 ± 0.8 | 168.3 ± 16.4 | 58 350 ± 9922 |
| 90/10 | 2.3 ± 0.4 | 0.6 ± 0.1 | 129.8 ± 1.3 | 129.4 ± 8.1 | 30 530 ± 12 091 |
| 70/30 | 2.9 ± 0.9 | 0.4 ± 0.1 | 130.4 ± 1.3 | 66.9 ± 8.9 | 15 333 ± 6890 |
| 50/50 | 2.4 ± 0.5 | 0.4 ± 0.1 | 123.6 ± 0.4 | 89.2 ± 10.3 | 9623 ± 9606 |
| PLLA/cellulose 1% nanosilica | | | | | |
| 100/0 | 5.7 ± 3.3 | 1.0 ± 0.1 | 128.6 ± 0.8 | 168.3 ± 16.4 | 58 350 ± 9922 |
| 90/10 | 2.3 ± 0.5 | 0.7 ± 0.1 | 108.7 ± 7.7 | 76.0 ± 14.0 | 41 473 ± 9204 |
| 70/30 | 2.2 ± 0.4 | 0.4 ± 0.1 | 25.0 ± 0.0 | 48.4 ± 20.8 | 41 165 ± 18 776 |
| 50/50 | 1.9 ± 0.5 | 0.3 ± 0.1 | 25.0 ± 0.0 | 135.1 ± 41.7 | 30 726 ± 11 732 |
| PDX/CA 1% nanosilica | | | | | |
| 100/0 | 1.4 ± 0.4 | 0.8 ± 0.1 | 32.1 ± 0.0 | 60.4 ± 8.4 | 50 485 ± 12 176 |
| 90/10 | 2.5 ± 0.9 | 1.0 ± 0.1 | 32.1 ± 0.0 | 68.1 ± 7.9 | 67 026 ± 15 791 |
| 60/40 | 2.5 ± 0.9 | 0.6 ± 0.1 | 32.1 ± 0.0 | 155.2 ± 5.0 | 35 036 ± 21 101 |

## 3.2. Data preparation, exploration and feature selection

A supervised learning problem is defined as inferring a functional mapping $y = f(x)$, based on a labelled training dataset $D = \{(x_1, y_1), (x_2, y_2), \ldots, (x_n, y_n)\}$. The present study is a regression problem: the inputs $x_i$

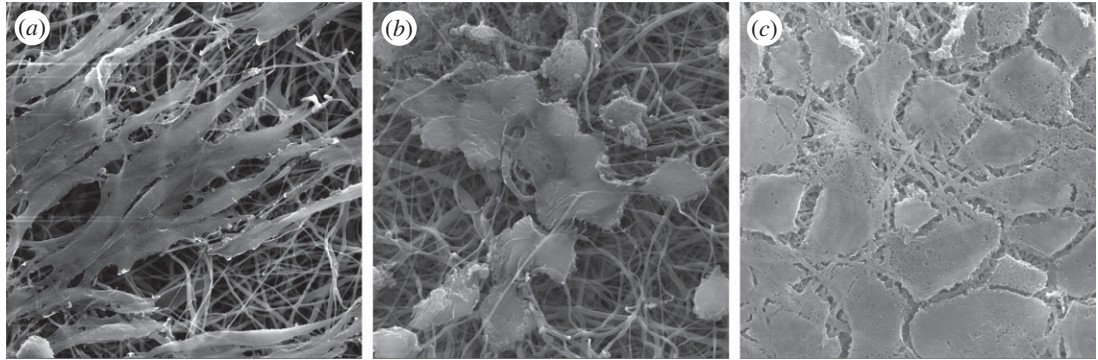

**Figure 1.** SEM images of L929 fibroblasts growing on 70/30 (*a*) PDX/KCG, (*b*) PDX/PSuc and (*c*) PDX/CA.

are d-dimensional vectors, $x_i = (x_i^1, x_i^2, \ldots, x_i^d) \in \mathbb{R}^d$ and the output variable $y$ takes continuous values ($y \in \mathbb{R}$). The goal of supervised learning algorithms is to analyse the training data and produce a function in order to predict the correct outcome for newly presented input data. The function used to connect input values/features $x$ to a predicted output $y$, is created by the ML model during the training phase. The obtained function is then tested by how accurately it performs on unseen data assumed to follow the same distribution as the training data [20].

Data exploration and feature selection were performed on the 13 scaffold families (table 2), representing a dataset of 182 observations and four features (pore diameter, fibre diameter, water contact angle and Young's modulus). Data were collected and regrouped to create an initial dataset *df* where each row is an observation (scaffold) and each column represents one feature (scaffold parameter) (electronic supplementary material, table S1). The initial dataset was loaded as a data frame and a variable $X$ was created, including the four scaffold parameters (fibre diameter, pore diameter, water contact angle and Young's modulus). A variable $y$ was also created and represented the target variable number of cells.

### 3.2.1. Low variance and high correlation filters

In ML, variances are calculated to make generalizations about a dataset, aiding in the understanding of data distribution. The variance of a variable is a measure of the average variation of values in the distribution with respect to the mean, given by

$$\sigma^2 = \frac{1}{N} \Sigma (X - \mu)^2, \tag{3.1}$$

where $\sigma^2$ is the variance, $N$ is the total number of observations, $X$ is the value of an individual observation and $\mu$ is the mean.

Low variance filter calculates the variance of each variable in our dataset and removes low variances (below a given threshold, set at $10^{-3}$ in our case) that would be of no significant contribution to the ML model. Variance calculations with scaled data was 0.017 for fibre diameter, 0.017 for pore diameter, 0.023 for Young's modulus and 0.149 for water contact angle.

High correlation filter calculates the correlation between independent scaled numerical variables. If the correlation between two variables is high (value closer to 1), it indicates that the variables have similar trends and are likely to carry similar information which can bring down the performance of some ML models. Figure 2 shows the Pearson correlation matrix performed on the four features (fibre diameter, pore diameter, water contact angle and Young's modulus). Coefficient correlations were between −0.19 and 0.42, i.e. no variable was highly correlated to another one.

### 3.2.2. Feature selection with random forest

Random forest is one of the most widely used algorithms for feature selection and allows us to compute the importance of each variable on the decision tree. The latter is a set of tests that are hierarchically organized and consists of nodes for testing features, edges to represent the outcome of each test and leaf/terminal nodes indicating the decision taken after computing all features. Random forest algorithm generates a large set of randomized decision trees to predict the target, and each feature's usage statistics are used to find the most informative subset of features in the dataset. Every decision tree has high variance, but when combined together and run in parallel, the resultant variance is low

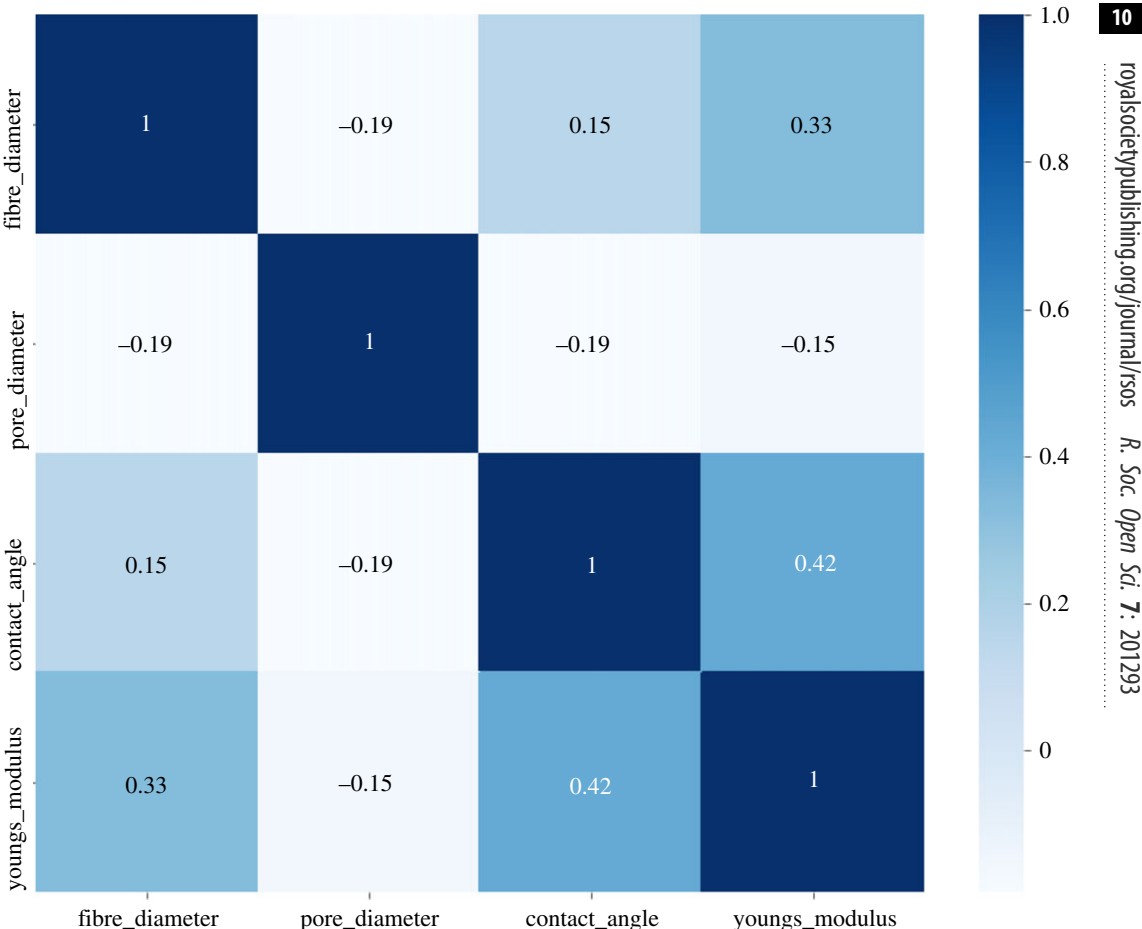

**Figure 2.** Pearson correlation matrix performed on four features: fibre diameter, pore diameter, water contact angle and Young's modulus.

as random forests train each individual decision tree independently on different bootstrapped samples of the training data and hence, the output does not depend on one decision tree but multiple decision trees. In the case of a regression problem, the predictions of each decision tree are averaged to make an overall prediction as final output (figure 3a). This technique is commonly known as bagging.

A score was calculated for each feature and the most predictive features were those with the highest scores. The feature importance graph showed that fibre diameter was the most important feature (0.39) while water contact angle and Young's modulus were the least important features (0.14 and 0.1, respectively) (figure 3b). This indicated that fibre diameter played an important role in dictating cell growth. The fibre surface is the first point of contact of cells as they first attach themselves to the fibre and exert the push–pull effect to be able to move and proliferate. Pore diameter, the second important feature, is responsible for cell penetration in scaffolds as well as the circulation of nutrients for cell growth. It is important to note that cells experience mechanical properties at the nanoscale and microscale, whereas mechanical properties of scaffolds are measured on the macroscale. Fibroblast cells are known to attach themselves on surfaces with water contact angle in the range of 60–80° [21]. This may explain the least impact exerted by this parameter as the scaffolds exhibited water contact angles in the range of 25–141.3°. The influence of this feature may vary with another cell type.

### 3.2.3. Recursive feature elimination and forward feature selection

RFE and FFS are applied to datasets with small number of input variables as is the case in this present study. RFE uses an accuracy metric to rank the features according to their importance. Linear regression was selected as model with four features and RFE gave '1' and 'True' for all four features. Three optimum features were found by the RFE method: fibre diameter, pore diameter and Young's modulus (score = 0.063). FFS is the opposite process of RFE as it tries to find the best features which can improve the performance of the model. In this study, the variables fibre diameter ($p = 0.003$) and pore diameter ($p = 0.004$) were retained as the most important parameters.

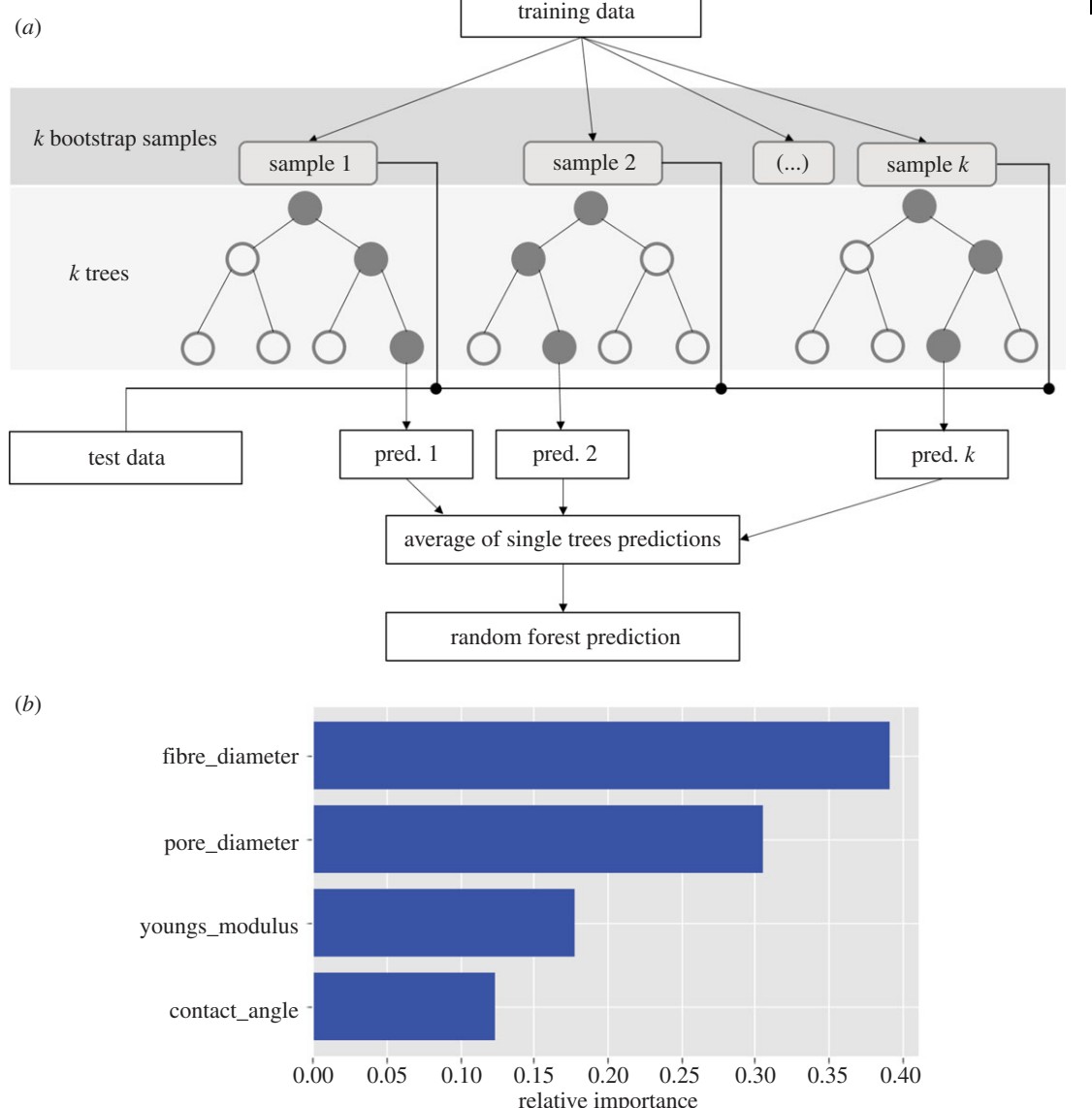

**Figure 3.** (*a*) Simplified structure of random forest regression algorithm. (*b*) Feature importance graph computed using random forest regressor, ranking the features based on their importance.

RFE and FFS scores confirmed the scores computed with random forest regressor. Among the three feature selection methods performed on the data, fibre diameter and pore diameter were identified as the most relevant features in the dataset. These two parameters could contribute most to the construction of our predictive model.

## 3.3. Evaluation and selection of the predictive model

After the training process, model testing is performed to determine the accuracy of each ML model. In general, for a selected ML model, the scores generated on the different test sets are not enough to establish a quantitative score to select and assess the robustness of the model. Hyperparameters are predefined parameters fixed before the training process. They can be tuned and optimized to achieve the maximal performance of a model. Thus, model selection is performed by maximizing the cross-validation score. Hyperparameters are tweaked in a principled way and a selection between different models is available.

Six regression models, namely linear regression, SVR, lasso regression, random forest regression, decision tree regression and k-NN regression, were employed to predict scaffold performance with *in vitro* cell culture data. Each ML algorithm was trained on an 80% random split of the dataset, while

**Table 3.** Model accuracy scores obtained after performing hyperparameter tuning with GridSearch.

| regression methods | accuracy (%) | MAPE (%) | Spearman correlation coefficient ($p$-value) |
| --- | --- | --- | --- |
| linear | 54.95 | 45.05 | 0.42 (**) |
| SVR | 55.63 | 44.37 | 0.42 (**) |
| lasso | 54.98 | 45.02 | 0.42 (**) |
| random forest | 62.74 | 37.26 | 0.64 (****) |
| decision tree | 53.91 | 46.09 | 0.39 (**) |
| k-NN | 54.46 | 45.44 | 0.42 (**) |

the remaining 20% of the data was used to test the cross-validated model and evaluate the accuracy of the ML algorithm. For the ML models to provide optimal prediction performance, hyperparameter tuning was performed with the GridSearchCV estimator. For each ML model, cross-validation scores were computed for all hyperparameter combinations to find the best one. While selecting a regression model, computing metrics like accuracy measures and error rates are important to evaluate the prediction accuracy of the model. The MAPE is the most common statistical measure used to forecast error and is calculated as the average of the absolute percentage errors of predictions. MAPE can be formalized by the following mathematical expression

$$\text{MAPE} = \frac{1}{n}\sum_{t=1}^{n}\left|\frac{A_t - P_t}{A_t}\right| \times 100, \tag{3.2}$$

where $n$ is the size of the sample, $P_t$ is the value predicted by the model for time point $t$ and $A_t$ is the value observed at time point $t$. The measure of the accuracy of the model is then calculated as accuracy = 100 − MAPE.

Table 3 shows the accuracy score, MAPE and Spearman correlation coefficient to the corresponding regression method after hyperparameter tuning. Random forest regression gave the best performance with a decent accuracy of 62.74% and a high degree of Spearman correlation (0.64) implying that the actual and predicted values had similar directional movement, i.e. when the actual values increased, the predicted values also increased. Compared to the other models, the random forest method has shown the lowest MAPE (37.26%), indicating a reasonable prediction of our model [22]. The following set of hyperparameters were adjusted for the random forest method: n_estimators (number of trees in the forests, set at 200), max_features (number of features to consider for splitting a node, set to sqrt(n_features)), max_depth (maximum number of levels in each decision tree, set at 40), min_samples_split (minimum number of samples required to split an internal node, set at 2), min_samples_leaf (minimum number of samples required to be at a leaf node, set at 4) and bootstrap (using bootstrap samples when building trees, set to True). The five other ML models yielded accuracy scores lower than 60% indicating that these statistical models were not presenting accurate mapping between the inputs and predicted output (table 3).

The correlation between the actual and predicted output data is another important factor when designing a predictive model. We evaluated the adequacy of the model by plotting the actual values from the test dataset against predicted values from the random forest model (figure 4). The plot displayed a statistically significant decent fit ($p < 3.7 \times 10^{-4}$), inferring a potential relationship between the predictors and the outcome that can be improved with more values closer to the fitted regression line. $R$-squared ($R^2$) represents the percentage of variance explained by covariates in the model and measures the proportion of variability in the outcome that has been explained by the model. It is ranged between zero and one. Usually, the higher the value, the better the model is able to explain the variability in the outcome, depending on the quality of the data. In our case, $R^2$ (0.3073) showed that more than 30% of the variance in the data is explained by the model. It is hypothesized that a better prediction accuracy will be seen by increasing the number of observations in the training set, and having more features in our dataset.

Fibre diameter values between 0.13 and 1.35 μm (including standard error of mean) were on/closest to the fitted line. An analysis of the experimental data indicated that these fibre diameters corresponded to the presence of biopolymers in the ratio range 10–30% in the polymeric blends with PDX, PHBV, PHB or PLLA (electronic supplementary material, table S2).

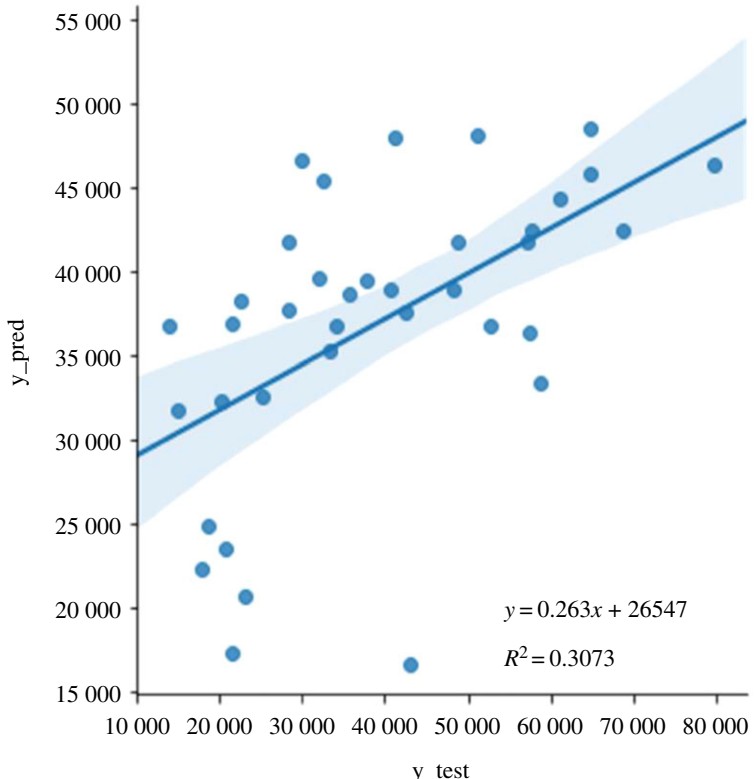

$$y = 0.263x + 26547$$

$$R^2 = 0.3073$$

**Figure 4.** Actual versus predicted plot showing the actual targets from the test dataset (y_test) against the predicted values by the random forest model (y_pred).

**Table 4.** *In vivo* estimation of number of blood vessels around explanted scaffolds.

| nanoscaffolds | fibre diameter (µm) | number of blood vessels (mm$^2$) | |
|---|---|---|---|
| | | two weeks | four weeks |
| PDX 100% | 1.28 | 20 ± 2 | — |
| PDX/CA 90/10% | 0.97 | 44 ± 2 | — |
| PDX/CA 60/40% | 0.64 | 247 ± 2 | 357 ± 3 |
| PLLA 100% | 0.97 | 298 ± 9 | 400 ± 14 |
| PLLA/cellulose 70/30% | 0.60 | 375 ± 7 | 444 ± 13 |
| PLLA/cellulose 50/50% | 0.28 | 409 ± 11 | 476 ± 10 |
| cellulose 100% | 0.25 | 449 ± 13 | 517 ± 6 |

## 3.4. Preliminary correlation with *in vivo* data: impact of fibre diameter on angiogenesis

The ML findings were correlated with *in vivo* biocompatibility experiments conducted on Wistar rats. Two families of scaffolds (PDX/CA and PLLA/cellulose) were implanted in the dorsal region of the rats and after two/four weeks, the scaffolds and surrounding tissues were removed and analysed. Angiogenesis is an important phenomenon in the wound healing process. The number of blood vessels was counted within a fixed distance of 100 µm from the explanted scaffold in a measured surface area (electronic supplementary material, figure S1). In this quantification study, a blood vessel was identified as one with a well-defined lumen consisting of red blood cells and an intact wall. The results showed a direct correlation between fibre diameter and angiogenesis whereby a decrease in fibre diameter resulted in an increase in number of blood vessels adjacent to the scaffolds, indicating enhanced angiogenesis (table 4).

# 4. Conclusion

We have shown for the first time the implications of incorporating ML methods to nanoscaffolds performance. A simplified predictive model was constructed and six regression algorithms were tested using the Seaborn/Scikit-learn Python toolkit. Hyperparameter tuning was performed to choose a set of optimal hyperparameters for each model in view of obtaining the best performance of each ML model, and random forest regression was found to be the model with the highest accuracy. Fibre diameter and pore diameter emerged as the two physico-chemical parameters which impacted more on the MTT values. Algorithms from transfer learning and reinforcement learning for increased accuracy are now being considered to increase the robustness of our model in predicting *in vitro* outcome. The final outcome would be a generalizable model that uses physico-chemical parameters of scaffolds as input and generates predictions of cells proliferation during *in vitro* cell culture in new scaffolds. The impact of different cell types on the model will also be studied.

Data accessibility. Provided as electronic supplementary material.

Authors' contributions. L.Y.S. carried out modelling studies, formal analysis and drafted the manuscript. N.G. carried out scaffolds and *in vitro* investigations, formal analysis, drafted the manuscript and helped in funding acquisition. I.C. and H.R. carried out scaffolds and *in vitro* investigations. F.G. carried out *in vivo* investigations. S.B. coordinated modelling studies and helped in funding acquisition. A.B.-L. conceptualized the project, coordinated the methodologies, funding acquisition (Principal Investigator) and all resources. A.B.-L. and S.B. participated in the writing—original draft preparation, writing-reviewing and editing. All authors gave final approval for publication and agree to be held accountable for the work performed therein.

Competing interests. We declare we have no competing interests.

Funding. This work was supported by the University of Mauritius—Pole of Innovation for Health Grant (K498) awarded by the Mauritius Research and Innovation Council (MRIC).

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
