## [Reviewer comments · Royal Society Open Science]

Review History

RSOS-201293.R0 (Original submission)

Review form: Reviewer 1

Is the manuscript scientifically sound in its present form?

Yes

Are the interpretations and conclusions justified by the results?

Yes

Is the language acceptable?

Yes

Do you have any ethical concerns with this paper?

No

Have you any concerns about statistical analyses in this paper?

No

Recommendation?

Accept with minor revision (please list in comments)

Comments to the Author(s)

This work focuses on engineering of a polymeric scaffold for tissue regeneration. Interestingly, it applies statistical methods to optimize the physicochemical properties of the scaffold using experimental in vitro data and physicochemical data of electrospun scaffolds tested for skin tissue engineering to model scaffold performance using machine learning. Correlation was sort between the physical properties and the MTT assay of L929 fibroblasts cells on the scaffolds. Six supervised learning algorithms were trained using Seaborn/Scikit-learn Python libraries. This work will add significant value to the readers as a first preliminary study on machine learning methods for the prediction of cell-material interactions on nanofibrous mats. Below are some minor revisions:

- 1) The language and grammar need to be revisited to improved readability
- 2) Sub-headings need to be made more descriptive
- 3) More detail to be provided on the quantities/composition of the polymers used in synthesizing the fibrous scaffolds

Decision letter (RSOS-201293.R0)

Dear Dr LUXIMON:

Title: Correlating in vitro performance with physico-chemical characteristics of nanofibrous scaffolds for skin tissue engineering using supervised machine l
Manuscript ID: RSOS-201293

Thank you for submitting the above manuscript to Royal Society Open Science. On behalf of the Editors and the Royal Society of Chemistry, I am pleased to inform you that your manuscript will be accepted for publication in Royal Society Open Science subject to minor revision in accordance with the referee suggestions. I apologise this has taken longer than usual. Please find the reviewers' comments at the end of this email.

The reviewers and handling editors have recommended publication, but also suggest some minor revisions to your manuscript. Therefore, I invite you to respond to the comments and revise your manuscript.

Because the schedule for publication is very tight, it is a condition of publication that you submit the revised version of your manuscript before 11-Nov-2020. Please note that the revision deadline will expire at 00.00am on this date. If you do not think you will be able to meet this date please let me know immediately.

Kind regards,
Dr Laura Smith
Publishing Editor, Journals

On behalf of the Subject Editor Professor Anthony Stace and the Associate Editor Professor Kim Jelfs.

RSC Associate Editor:
Comments to the Author:
Please meet the final requests of the referee.

RSC Subject Editor:
Comments to the Author:

(There are no comments.)

Reviewer comments to Author:

Reviewer: 1

Comments to the Author(s)

This work focuses on engineering of a polymeric scaffold for tissue regeneration. Interestingly, it applies statistical methods to optimize the physicochemical properties of the scaffold using experimental in vitro data and physicochemical data of electrospun scaffolds tested for skin tissue engineering to model scaffold performance using machine learning. Correlation was sort between the physical properties and the MTT assay of L929 fibroblasts cells on the scaffolds. Six supervised learning algorithms were trained using Seaborn/Scikit-learn Python libraries. This work will add significant value to the readers as a first preliminary study on machine learning methods for the prediction of cell-material interactions on nanofibrous mats. Below are some minor revisions:

- 1) The language and grammar need to be revisited to improved readability
- 2) Sub-headings need to be made more descriptive
- 3) More detail to be provided on the quantities/composition of the polymers used in synthesizing the fibrous scaffolds

Author's Response to Decision Letter for (RSOS-201293.R0)

See Appendix A.

Decision letter (RSOS-201293.R1)

Dear Dr LUXIMON:

Title: Correlating in vitro performance with physico-chemical characteristics of nanofibrous scaffolds for skin tissue engineering using supervised machine l
Manuscript ID: RSOS-201293.R1

It is a pleasure to accept your manuscript in its current form for publication in Royal Society Open Science. The chemistry content of Royal Society Open Science is published in collaboration with the Royal Society of Chemistry.

On behalf of the Subject Editor Professor Anthony Stace and the Associate Editor Professor Kim Jelfs.

RSC Associate Editor
Comments to the Author:
(There are no comments.)

Reviewer(s)' Comments to Author:

Appendix A

Response to reviewer comments

1) The language and grammar need to be revisited to improved readability.

We have thoroughly revised the manuscript for language and grammar.

2) Sub-headings need to be made more descriptive.

Sub-headings have been modified as indicated in red.

3) More detail to be provided on the quantities/composition of the polymers used in synthesizing the fibrous scaffolds.

More details have been added in the experimental section in red.